# Assessment of Cervical IL-6 Levels and Neonatal Inflammatory Response in Preterm Birth Following Preterm Premature Rupture of Membranes

**DOI:** 10.3390/cimb47100838

**Published:** 2025-10-12

**Authors:** Gusztav Labossa, Tamas Koszegi, Balint Farkas, Bernadett Nagy, Rita Jakabfi-Csepregi, Nelli Farkas, Kalman Kovacs

**Affiliations:** 1Department of Obstetrics and Gynecology, University of Pecs Medical School, 7624 Pecs, Hungary; labossa.gusztav@pte.hu (G.L.); nagy.bernadett@pte.hu (B.N.); kovacs.kalman@pte.hu (K.K.); 2Szentagothai Research Centre, University of Pecs, 7624 Pecs, Hungary; tamas.koszegi@aok.pte.hu (T.K.); ritacsepregi93@gmail.com (R.J.-C.); 3National Laboratory on Human Reproduction, University of Pecs, 7624 Pecs, Hungary; 4Institute of Bioanalytics, University of Pecs Medical School, 7624 Pecs, Hungary; nelli.farkas@aok.pte.hu

**Keywords:** cervical fluid interleukin-6, inflammatory markers, neonatal inflammatory response, preterm birth, preterm delivery, preterm premature rupture of membranes

## Abstract

Background: Preterm premature rupture of membranes (PPROM) is a leading cause of preterm delivery, occurring in 40–50% of cases, with a 3–4% overall incidence. During expectant management, chorioamnionitis is typically monitored using serum inflammatory markers (e.g., leukocyte count, CRP), though cervical interleukin-6 (IL-6) has emerged as a promising local marker. This study investigated the correlation between cervical IL-6 and maternal and neonatal inflammatory parameters. Methods: This prospective non-randomized clinical trial was performed with 51 patients with expectantly managed PPROM. Samples were obtained twice a week using cervical swabs placed into a stabilizer solution. Cervical IL-6 levels were measured by routine automated chemiluminescence immunoassay, with reference to albumin levels. Maternal serum C-reactive protein (CRP) levels and leukocyte counts and neonatal serum CRP and procalcitonin (PCT) levels were also determined. Spearman correlations of the IL-6 level with other markers and clinicopathological parameters were examined. Results: Cervical IL-6 levels correlated more strongly with neonatal CRP and PCT levels on the first day after delivery than with maternal serum markers while showing no significant association with the PPROM-to-delivery interval. Conclusion: Cervical IL-6 level determination may help to inform the timing of labor induction in cases of PPROM, with consideration of the maternal and neonatal conditions. We believe that the monitoring of the cervical IL-6 level could enable good prediction of perinatal inflammation, although more data are needed to support this hypothesis.

## 1. Introduction

Preterm birth is a global health issue, affecting almost 15 million newborns worldwide [1] and causing almost one-third of all neonatal deaths [2]. It is associated with short-term complications (respiratory distress syndrome, bronchopulmonary dysplasia, necrotizing enterocolitis, and intraventricular hemorrhage) and long-term adverse health events (cerebral palsy, behavioral and cognitive disorders, and mental health conditions) [3,4,5]. Preterm premature rupture of membranes (PPROM) remains a major cause of preterm delivery, involved in 40–50% of cases, with an overall incidence of 3–4% (0.4–0.7% before 28 weeks of gestation) [6,7,8]. Especially in the second trimester, PPROM is associated with high mortality rates. Although the pathophysiology underlying preterm delivery is understood to involve systemic, uterine, fetal, and uteroplacental factors, it is also associated with invasive intrauterine interventions, chorioamnionitis, and oxidative stress processes [9]. The consequences of preterm delivery, including low birth weight, intracranial hemorrhage, periventricular leukomalacia, necrotizing enterocolitis, and psychomotor developmental disorders, have negative impacts on neonatal morbidity and mortality [10].

Intra-amniotic infection is present in about 70% of all PPROM cases, as confirmed by amniotic fluid culture studies [4,11,12,13]. The likelihood of maternal inflammation development or intensification increases over time in PPROM. The management of PPROM demands the achievement of an appropriate balance between the advantages of prolonged intrauterine development and the risk of infection development. Close monitoring [i.e., of the maternal body temperature; fetal heart rate; C-reactive protein (CRP), interleukin-6 (IL-6), and procalcitonin (PCT) levels; white blood cell (WBC) count; and amniotic fluid index] is required to detect chorioamnionitis symptoms and minimize the risk of neonatal and maternal complications [14,15]. The monitoring of cervically acquired inflammatory markers such as IL-6 has been reported to be reliable for the assessment of chorioamnionitis [16].

With the expectant management of PPROM, which aims to prolong intrauterine fetal development to increase the birth weight and gestational age and enhance the functional maturity of the lungs, pregnancies need to be terminated to increase the chance of neonatal survival when signs of fetal distress (evidenced by cardiotocography and ultrasound) or intrauterine infection are detected; obstetrical decision making is thus the main challenge of this approach. Current guidelines for PPROM stipulate that pregnancies should be terminated when increases in maternal inflammatory parameters indicate the progression of inflammation, thereby endangering the health or, in severe cases, the life of the mother [17]. In these instances, however, the development of more severe infection in neonates is inevitable. In this study, we aimed to identify a method for the more precise determination of the timing of pregnancy termination in PPROM, to contribute to the delivery of healthier neonates and their more rapid recovery. Although previous studies have already proposed measuring IL-6 in amniotic fluid as well as in cervical or vaginal samples [16,18,19], these investigations have not become widespread and often did not include a neonatal focus. To meet this goal, we examined correlations of local (cervical) IL-6 levels with maternal and neonatal serum inflammatory markers. We hypothesize that cervical interleukin-6 (IL-6) concentrations, assessed during the expectant management of PPROM, exhibit a stronger correlation with early neonatal inflammatory markers (CRP and PCT) than with maternal serum parameters.

## 2. Materials and Methods

### 2.1. Study Design and Participants

This prospective, observational cohort study was performed with 51 pregnant women hospitalized for PPROM between gestational weeks 24 and 36 who received conservative (expectant) treatment to avoid preterm delivery at the University of Pecs Clinical Center’s Department of Obstetrics and Gynecology between September 2021 and March 2023. Demographic data (age, parity) were collected. Treatment included antenatal steroid and antibiotic prophylaxis on admission and 48 h of intravenous tocolysis after admission. Only severe acute respiratory syndrome coronavirus 2-negative women with singleton pregnancies were included. Patients with histories of hypertension; pre-eclampsia; diabetes; liver, heart, or kidney disease; and fetal malformation were excluded. All participants provided written informed consent, and the University of Pecs Institutional Ethical Review Board approved the study (5643-PTE 2023).

### 2.2. Sample Collection and Processing

Maternal peripheral blood and cervical samples were collected on admission and twice a week (on Tuesdays and Thursdays) thereafter. Neonatal peripheral blood samples were collected on the first and second days of life. Cervical samples were collected with Dacron polyester swabs (Fisherbrand™ Pittsburg, PA, USA) after vaginal speculum placement. The tips of sample-containing swabs were separated and submerged into a 15 mL Falcon tube (Sigma-Aldrich, Burlington, MA, USA) containing 10 µg/mL aprotinin protease inhibitor cocktail containing isotonic phosphate-buffered saline (PBS pH 7.2) for transportation. The tubes were vortexed for 4 min, and the released cervical fluid–PBS–aprotinin samples were pipetted into Eppendorf centrifuge tubes. The extracts were centrifuged at 1500× *g* for 5 min at 8 °C, and the supernatants were collected for analysis. All samples were processed and analyzed in the fully accredited Department of Laboratory Medicine (NAH-9-0008/2021) using standard, routine automated methods. The supernates of the centrifuged cervical extracts were tested immediately. IL-6 levels were determined by electro-chemiluminescence immunoassay (Cobas e411; Roche, Mannheim, Germany), and albumin concentrations were determined by immune turbidimetry using a Cobas 8000 automated chemical analyzer and a microalbumin assay procedure (Roche, Mannheim, Germany). IL-6/albumin ratios were calculated. Serum WBC counts were measured by a Sysmex 9000 fully automated hematology analyzer (Sysmex Hungária Ltd., Budapest, Hungary), and high-sensitivity CRP levels were determined from maternal blood samples on the Cobas 8000 analyzer. Serum CRP and PCT levels (Roche, e411) were determined from neonatal blood samples. In order to obtain the results as soon as possible, the swabs were not weighed and/or frozen, as published in other studies [20,21]. The IL-6 levels were therefore referred to albumin, and the whole procedure gave results within three hours from sample collection.

### 2.3. Statistical Analysis

Statistical analyses were performed with the R statistical software package (version 4.1) (R Core Team (2021). R: A language and environment for statistical computing (R Foundation for Statistical Computing, Vienna, Austria)). Continuous data are presented as means with standard deviations, and categorical data are presented as counts and percentages. To examine correlations between variables, Spearman’s rank correlation analysis was performed. Specifically, we investigated correlations of the cervical IL-6/albumin ratios with the PPROM–delivery and admission–delivery intervals and with the neonatal serum CRP and PCT levels. We also examined correlations between individual and combined maternal and neonatal inflammatory blood parameters; the examination of multiple parameters together enables the more confident determination of the undergoing pathological processes. The significance level was set to *p* < 0.05.

## 3. Results

### 3.1. Demographic and Obstetric Characteristics

The mean maternal age was 31 ± 1.32 (range, 22–45) years. Of the 51 women included, 31.3% (n = 16) were nulliparous, 39.6% (n = 19) had delivered one child, 18.8% (n = 9) had delivered two children, 8% (n = 4) had delivered three children, 4.2% (n = 2) had delivered four children, and 2% (n = 1) had delivered five children (Table 1). The mean gestational age at the time of PPROM diagnosis was 111.9 days (28 weeks), and all participants delivered at preterm (mean gestational age 32.3; min.: 24; max.: 36). None of the participants tested prepartum positively for any sexually transmitted diseases (*N. Gonorrhea*, *C. Trachomatis*, *T. Vaginalis*), although by admission, 24% (12/51) of the study participants tested positive for genital tract infections (five for *Streptococcus agalactiae*, one for *Staphylococcus aureus*, six for *Candida albicans*).

### 3.2. Correlations and Dynamics of IL-6 Levels

Our results indicate that cervical IL-6 levels expressed as IL-6/albumin ratios are more strongly correlated with neonatal inflammatory markers than maternal CRP or WBC values, suggesting a higher predictive value for neonatal inflammation in PPROM. The final predelivery cervical IL-6/albumin levels correlated with neonatal serum CRP and PCT levels measured on the first day after delivery (r = 0.61 and 0.62, respectively; both *p* < 0.05; Figure 1). Weaker correlations were observed between the final predelivery maternal serum CRP level and WBC count and first-day neonatal serum CRP and PCT levels. As the majority of neonates received various treatments (e.g., antibiotics, oxygen, fluid therapy) immediately after birth and did not respond uniformly to these treatments, we did not consider the examination of second-day neonatal laboratory values to be justified. The combined examination of the maternal CRP level and WBC count and the neonatal CRP and PCT levels showed that inflammation may affect neonates’ conditions at birth before maternal inflammatory laboratory values indicate the worsening of mothers’ conditions. These results indicate that IL-6, as a local inflammatory factor, might be a potentially good predictor of the neonatal condition. The cervical IL-6 level did not correlate with the PPROM–delivery or admission–delivery interval.

Interestingly, an elevated IL-6/albumin ratio (>2) was observed in 83% (5/6) of patients with positive cervical bacterial cultures. Furthermore, 60% (3/5) of their neonates subsequently developed neonatal sepsis. Cervical samples were collected on admission and twice weekly thereafter. IL-6 concentrations from the first and last predelivery measurements for each participant are shown in Figure 2. In most cases, levels increased as delivery approached, highlighting IL-6′s role as an early indicator of intrauterine inflammation.

## 4. Discussion

Numerous studies have been conducted to explore the use of markers such as IL-6 in addition to traditional inflammatory markers for monitoring [22,23,24,25]. Almost without exception, these studies have focused on the maternal condition and the identification of the marker that most rapidly and sensitively indicates the onset and progression of the inflammatory process. Studies of newer inflammatory markers have not extended to the postdelivery neonatal condition or correlations thereof with maternal blood and local or neonatal blood markers.

With the expectant management of PPROM, neonates’ peripartum condition has been found to depend on the gestational age at the time of membrane rupture and the latency period of medical treatment. Rates of complications (e.g., respiratory distress syndrome, intrauterine hypoxia) are significantly lower when PPROM occurs in gestational weeks 28–34 than when it occurs before gestational week 28 [25]. Other studies have revealed no significant difference in the incidence of hemoculture-confirmed neonatal sepsis or the survival rate according to whether PPROM is managed expectantly or actively. Secondary findings have shown no increase in intrauterine or perinatal mortality but a higher rate of cesarean delivery. Prophylactic antibiotic treatment has been confirmed to decrease neonatal and maternal infection rates [26,27].

The previous literature has emphasized the diagnostic value of maternal serum inflammatory markers in assessing intrauterine infection but with limitations in predicting neonatal outcomes. McElrath highlights the clinical dilemma in balancing the prolongation of gestation against the risk of infection in PPROM cases near the limit of viability, underscoring the need for more precise indicators [28].

Our findings reinforce the growing body of evidence that local inflammatory markers, particularly cervical IL-6, may offer earlier and more direct insight into fetal inflammatory responses than systemic maternal biomarkers. This aligns with the current understanding of the pathophysiology of spontaneous preterm birth (sPTB), in which inflammation plays a central role. Intra-amniotic infection and sterile inflammation activate the maternal–fetal inflammatory axis long before overt clinical signs of chorioamnionitis emerge [29]. Our findings suggest that cervical IL-6 measurement could become an integral component of individualized PTB risk assessment, particularly in pregnancies complicated by PPROM.

The combination of localized inflammatory markers with advanced biosignal monitoring in the neonatal period may offer a new, integrated pathway for earlier detection and more targeted interventions. Lungu et al. demonstrated that the use of multimodal biosignal integration—combining NIRS, pulse oximetry, and skin temperature monitoring—can predict neonatal sepsis up to 48 h in advance with high accuracy [30]. Our findings suggest that cervical IL-6 could be used prenatally as a complementary predictive marker.

### Strengths and Limitations

The main strength of this study is its novel focus on the neonatal, rather than maternal, condition. Notable limitations are the relatively small number of cases and lack of subgroup analysis according to specific obstetric pathologies that could significantly influence the neonates’ condition. We recommend the performance of such subgroup analyses in future studies. In addition, samples were collected only twice a week. The twice-a-week sampling schedule was chosen to minimize the burden on pregnant women and to avoid increasing the risk of potential complications or infections. Moreover, appropriate cut-off values were difficult to determine. Musilova et al. [18,19] recommended a cut-off value of ≥3000 pg/mL (≥3 ng/mL) in amniotic fluid and around 350 pg/mL in cervical and vaginal fluid. A technical limitation in determining a cut-off value is the choice of a reliable reference that can be obtained at the bedside. Most studies weigh the swabs and calculate the interleukin levels with reference to the sample weight. However, in clinical practice, this is not feasible, and the protein content of the swab extracts may vary within a wide range. Moreover, the delayed determination of the interleukin levels may not help with proper decision making. Therefore, we chose the measurement of albumin as a reference from the cervical samples, speeding up the lab procedure to a large extent. This study’s other limitations are the single-center design, limited sample size, sampling frequency, and the restriction of neonatal outcomes to laboratory markers. Future studies with larger cohorts, multicenter settings, more frequent sampling, and long-term follow-up of neonates could help validate and extend these findings.

## 5. Conclusions

Our findings suggest that cervical interleukin-6 (IL-6) levels, measured during the expectant management of PPROM, correlate more strongly with early neonatal inflammatory markers (CRP and PCT) than with maternal serum indicators. We introduced the determination of cervical albumin as a reference compound with a fast automated method that speeds up the analysis of individually collected samples with a turnaround time of no more than 3 h. This supports the utility of IL-6 as a localized biomarker of fetal inflammatory status, potentially aiding in timely decision making regarding delivery. Cervical IL-6 measurement can be a useful monitoring method in the context of the expectant management of PPROM. It aids timely decision making about pregnancy termination with the consideration of the maternal and neonatal conditions, reducing neonatal morbidity and mortality.

Public health benefits include the potential to reduce neonatal morbidity and mortality by minimizing diagnostic delays and avoiding overtreatment. Our approach prioritizes fetal outcomes without compromising maternal safety, making it a promising tool in perinatal care optimization. Ultimately, this research supports the development of predictive protocols that are less reliant on non-specific maternal symptoms and more attuned to the pathophysiological processes occurring at the fetal–maternal interface. Future multicenter studies with larger cohorts are warranted to validate these findings and explore their implementation in clinical protocols.

## Figures and Tables

**Figure 1 cimb-47-00838-f001:**
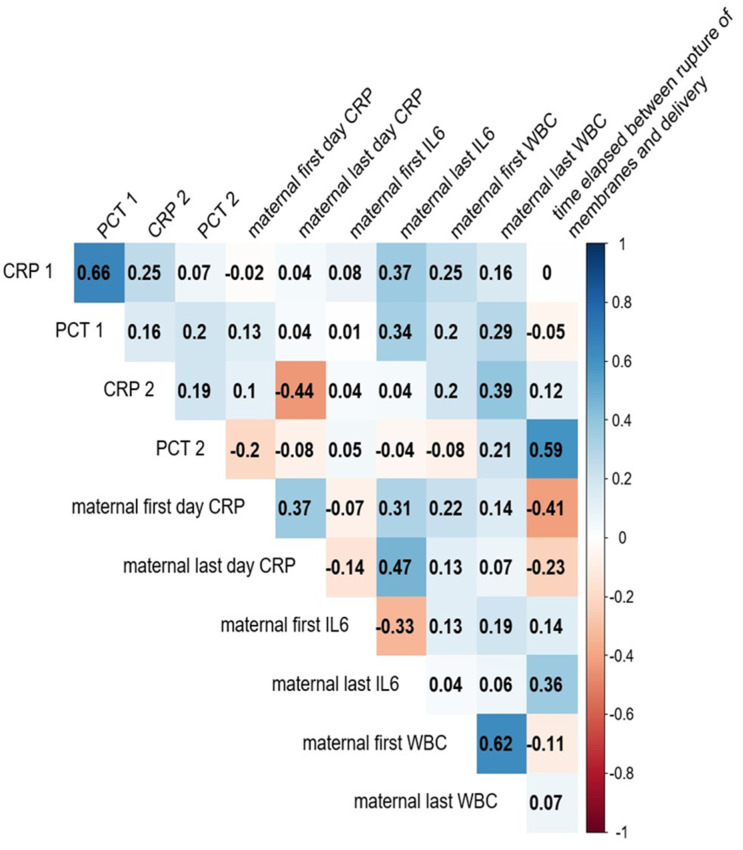
Spearman correlation matrix illustrating the associations between maternal inflammatory markers (horizontal axis) and neonatal inflammatory markers (vertical axis). IL-6 data are taken as IL-6/albumin ratios. Positive correlations are indicated in blue, while negative correlations are shown in red. Stronger correlations are represented by darker shades, highlighting potential maternal–neonatal inflammatory linkages.

**Figure 2 cimb-47-00838-f002:**
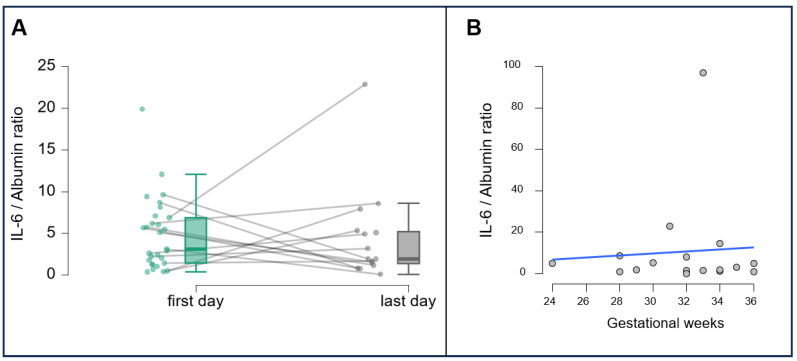
(**A**) Changes in cervical IL-6/albumin ratio (concentrations IL-6 pg/mL, albumin mg/L) measured at hospital admission (first day) and immediately before delivery (last day) in women with PPROM. Each line represents one patient, illustrating individual trends over the latency period. (**B**) Scatter plot showing individual IL-6/albumin ratio (concentrations IL-6 pg/mL, albumin mg/L) measured at the last time point across gestational weeks (24–36). The blue line represents the fitted linear regression model—no statistically significant association was found between gestational age and IL-6 levels.

**Table 1 cimb-47-00838-t001:** Summary of the characteristics of the study population.

Characteristics	Urban (n/%)	Rural (n/%)	Anemia (HGB < 120 g/L; n/%)	UTI (n/%)	GTI (n/%)	TA-AC (n/%)	Cerclage (n/%)	Spontaneous Delivery (n/%)	Cesarean Delivery (n/%)
**Environment of origin**	30/59	21/41							
**Pregnancy-related complications**			41/80	0	12/24				
**Invasive interventions during pregnancy**						3/6	1/2		
**Way of delivery**								15/29	36/71

UTI (urinary tract infection); GTI (genital tract infection); TA-AC (transabdominal amniocentesis). Anemia was determined by serum hemoglobin levels below 120 g/L.

## Data Availability

Data collected during this study is available on request from the corresponding author.

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
