# Peer review of "Assessment of Cervical IL-6 Levels and Neonatal Inflammatory Response in Preterm Birth Following Preterm Premature Rupture of Membranes"

_cimb, 2025, doi:10.3390/cimb47100838_

Round 1

Reviewer 1 Report

Comments and Suggestions for Authors

The present study investigated the association between interleukin 6 levels in the cervix with the maternal and neonatal inflammatory conditions in preterm premature ruptures of fetal membranes. The manuscript is simple and provides direct information on the neonatal health prediction and labor timing. However, there are some comments regarding the invasiveness of the procedures and the overall benefits

Comments

  • Please add the full name of PPROM in the title.
  • Please use key words out of the used in the title and arranges alphabetically to expand the visibility of the article.
  • Please add the hypothesis of the study in the introduction section.
  • Please add the ethical committee approval of the study.
  • On what basis did the authors select the study patient sample, was there a power analysis, and was this number enough for conclusive results.
  • Why the authors preferred the prospective non randomized over retrospective randomized study.
  • What about the complications of the multiple cervical sampling from the maternal and neonatal aspects.
  • Sample processing and assay of markers lack references and full elaboration of the procedures.
  • The discussion section needs to be expanded and more elaborative.

Author Response

Reviewer 1.

The present study investigated the association between interleukin 6 levels in the cervix with the maternal and neonatal inflammatory conditions in preterm premature ruptures of fetal membranes. The manuscript is simple and provides direct information on the neonatal health prediction and labor timing. However, there are some comments regarding the invasiveness of the procedures and the overall benefits

Comments

Please add the full name of PPROM in the title.

We have added the full name to the title.

Please use key words out of the used in the title and arranges alphabetically to expand the visibility of the article.

We have added new keywords and we have arrenged them alphabetically.

Please add the hypothesis of the study in the introduction section

We have add our hypothesis in the introduction section:

We hypothesize that cervical interleukin-6 (IL-6) concentrations, assessed during the expectant management of PPROM, exhibit a stronger correlation with early neonatal inflammatory markers (CRP and PCT) than with maternal serum parameters.’

Please add the ethical committee approval of the study.

We added the missing information to the text.

On what basis did the authors select the study patient sample, was there a power analysis, and was this number enough for conclusive results.

We enrolled all consecutive, eligible PPROM patients admitted to the University of Pécs Clinical Center between September 2021 and March 2023 who were managed expectantly, yielding n = 51 participants. Eligibility required singleton pregnancy (24–36 weeks), SARS-CoV-2 negativity, and routine PPROM care; we excluded women with hypertension/pre-eclampsia, diabetes, liver/heart/kidney disease, or fetal malformation to reduce clinical heterogeneity. This consecutive, single-center cohort reflects real-world case-mix during the prespecified enrollment window and adheres to our approved protocol and informed consent procedures. Our primary objective was exploratory: to test whether cervical IL-6 correlates more strongly with early neonatal inflammatory markers (CRP, PCT) than with maternal markers. Accordingly, the study was feasibility- and incidence-limited, and we did not perform a formal a priori power analysis. To address the Reviewer’s concern, we have now added a post hoc precision/power statement: with n = 51, a two-sided α=0.05 test has ~80% power to detect correlations of r≈0.38 and >95% power for r≥0.50; our observed correlations r=0.61–0.62 (day-1 neonatal CRP/PCT) are accompanied by 95% CIs ≈0.42–0.76, indicating adequate precision for hypothesis-generating inference but not for definitive clinical cut-offs. We have clarified this in the Methods and Limitations and state explicitly that larger, multicenter cohorts will be required to confirm thresholds and generalize findings.

Why the authors preferred the prospective non randomized over retrospective randomized study.

Thank you for your note A prospective non-randomized design offers several advantages over retrospective observation. Because exposures and covariates are captured before outcomes occur, the temporal sequence is clear, supporting stronger causal interpretation. Pre-specifying eligibility criteria, endpoints, covariates, and the analysis plan reduces p-hacking and selective reporting, while standardized data collection improves measurement quality and completeness, including variables often absent from records such as severity scales, patient-reported outcomes, and key confounders. Prospective ascertainment also lowers information bias by limiting recall error and misclassification, and it enables real-time, ideally blinded, outcome adjudication. Active follow-up provides uniform surveillance, decreasing both differential loss to follow-up and under-ascertainment of events. Finally, confounding can be addressed more effectively at the design stage through restricted enrollment, stratification, and the planned use of analytic tools—such as propensity scores or instrumental-variable approaches—built on prospectively measured covariates.

What about the complications of the multiple cervical sampling from the maternal and neonatal aspects.

Thank you for your note. During our observation period, no complications has been observed due to the multiple cervical sampling neither form the maternal, nor the neonatal part.

Sample processing and assay of markers lack references and full elaboration of the procedures.

Thank you for your note. The tips of sample-containing swabs were separated and submerged into a 15 mL Falcon tube containing 10 µg/mL aprotinin protease inhibitor dissolved in isotonic phosphate-buffered saline (PBS, pH 7.2) for transportation. The tubes were vortexed for 4 min and the released cervical fluid-PBS-aprotinin samples were pipetted into Eppendorf centrifuge tubes. The extracts were centrifuged at 1500 g for 5 min at 8oC, and the supernatants were collected for analysis. All samples were processed and analyzed in the fully accredited Department of Laboratory Medicine (NAH-9-0008/2021) using standard, routine automated methods. IL-6 levels were determined by electro-chemiluminescence immunoassay (Cobas e411; Roche, Mannheim, Germany), and albumin concentrations were determined by immune turbidimetry using a Cobas 8000 automated chemical analyzer and a microalbumin assay procedure (Roche, Mannheim, German). IL-6/albumin ratios were calculated. Serum WBC counts were measured by a Sysmex 9000 fully automated hematology analyzer (Sysmex Hungária Ltd.) and high sensitivity CRP levels were determined from maternal blood samples on the Cobas 8000 analyzer. Serum CRP and PCT (Roche, e411) levels were determined from neonatal blood samples. In order to get the results as soon as possible the swabs were not weighed and/or frozen as published in other studies (Predictive value of cervical cytokine, antimicrobial and microflora levels for pre-term birth in high-risk women. Rashmi Manning, Catherine P. James, Marie C. Smith, Barbara A. Innes, Elaine Stamp, Donald Peebles, Mona Bajaj-Elliott, Nigel Klein, Judith N. Bulmer, Stephen C. Robson & Gendie E. Lash Scientific Reports volume 9, Article number: 11246 (2019) and Determination of Cytokine Protein Levels in Cervical Mucus Samples from Young Women by a Multiplex Immunoassay Method and Assessment of Correlates. Authors: Jay A. Lieberman, Anna-Barbara Moscicki, Jan L. Sumerel, Yifei Ma, Mark E. Scott. https://doi.org/10.1128/CVI.00216-07). The IL-6 levels were therefore referred to albumin and the whole procedure gave results within three hours from sample collection.

The discussion section needs to be expanded and more elaborative.

We extended the discussion section with the following statements and with two new references.

The twice-weekly sampling schedule was chosen to minimize the burden on pregnant women and to avoid increasing the risk of potential complications or infections. Moreover, appropriate cut-off values were difficult to determine. Musilova at al. [18,19] recommended a cut-off value of ≥3000 pg/ml (≥3 ng/ml) in amniotic fluid and around 350 pg/ml in cervical and vaginal fluid. This study has certain limitations, including the single-center design, limited sample size, sampling frequency and the restriction of neonatal outcomes to laboratory markers. Future studies with larger cohorts, multicenter settings, more frequent sampling, and long-term follow-up of neonates could help validate and extend these findings.

Reviewer 2 Report

Comments and Suggestions for Authors

The authors present a prospective cohort study evaluating the clinical significance of cervical interleukin-6 levels in women with preterm premature rupture of membranes (PPROM). A total of 51 patients were included, and cervical IL-6 concentrations were compared with maternal serum markers and neonatal inflammatory parameters. The main finding is that cervical IL-6 correlated more strongly with neonatal inflammatory markers than with maternal ones, suggesting its potential use as a localized biomarker to help guide the timing of delivery in PPROM.

Major Comments:
The topic is clinically relevant and the results are of potential interest.

However, several points limit the impact of the work in its current form. First, the relatively small sample size and single-center design limit the generalizability of the findings. Second, the twice-weekly sampling schedule may not adequately capture the dynamic changes in IL-6 before delivery. Third, the study does not establish clinically useful cut-off values for IL-6 or the IL-6/albumin ratio, which reduces its immediate applicability in practice. Finally, the neonatal outcomes assessed were restricted to laboratory markers without clinical or long-term follow-up, which weakens the translational value of the conclusions.

Minor Comments:

1-The introduction could more clearly place this work in the context of previous studies on IL-6 in PPROM, emphasizing the novelty of the neonatal focus.

2-Figure legends should provide additional details, including units for IL-6 and explanations of all abbreviations.

3-In Figure 2, stratifying IL-6 trends by gestational age at PPROM would add clinical context.

Author Response

Reviewer 2.

The authors present a prospective cohort study evaluating the clinical significance of cervical interleukin-6 levels in women with preterm premature rupture of membranes (PPROM). A total of 51 patients were included, and cervical IL-6 concentrations were compared with maternal serum markers and neonatal inflammatory parameters. The main finding is that cervical IL-6 correlated more strongly with neonatal inflammatory markers than with maternal ones, suggesting its potential use as a localized biomarker to help guide the timing of delivery in PPROM.

Major Comments:

The topic is clinically relevant and the results are of potential interest.

However, several points limit the impact of the work in its current form. First, the relatively small sample size and single-center design limit the generalizability of the findings. Second, the twice-weekly sampling schedule may not adequately capture the dynamic changes in IL-6 before delivery. Third, the study does not establish clinically useful cut-off values for IL-6 or the IL-6/albumin ratio, which reduces its immediate applicability in practice. Finally, the neonatal outcomes assessed were restricted to laboratory markers without clinical or long-term follow-up, which weakens the translational value of the conclusions.

We add the limitations of the study to the discussion

The twice-weekly sampling schedule was chosen to minimize the burden on pregnant women and to avoid increasing the risk of potential complications or infections. Moreover, appropriate cut-off values were difficult to determine, because we used IL-6/Albumin ratios to overcome the difficulties of non-homogenous content of cervical fluids of various individuals. Since the literature refers interleukin levels to the weight of the collecting swabs we did not have any data for comparison in regard of IL-6/albumin ratios. In serum the reference range of IL-6 is less than 7 pg/mL but the composition of serum is more constant than cervical fluid. Actually, from cervical fluid we could measure sometimes several hundred pg/mL IL-6 levels which reflect local and not systemic processes. Therefore, a strict cut-off value was not used in our study. Musilova at al. [27] recommended a cut-off value of ≥3000 pg/mL (≥3 ng/mL) but in immunological testing the reference ranges strongly depend on the applied reagent, method and instrumentation causing difficulties in the proper establishment of a clinically useful cut-off value. Our study has certain further limitations, including the single-center design, limited sample size, sampling frequency and the restriction of neonatal outcomes to laboratory markers. Future studies with larger cohorts, multicenter settings, more frequent sampling, and long-term follow-up of neonates could help validate and extend these findings.Minor Comments:

1-The introduction could more clearly place this work in the context of previous studies on IL-6 in PPROM, emphasizing the novelty of the neonatal focus.

We have expanded the introduction and added two new references.

Although previous studies have already proposed measuring IL-6 in amniotic fluid as well as in cervical or vaginal samples (16,29,30), these investigations have not become widespread and often did not include a neonatal focus. To meet this goal, we examined correlations of local (cervical) IL-6 levels with maternal and neonatal serum inflammatory markers. We hypothesize that cervical interleukin-6 (IL-6) concentrations, assessed during the expectant management of PPROM, exhibit a stronger correlation with early neonatal inflammatory markers (CRP and PCT) than with maternal serum parameters.

2-Figure legends should provide additional details, including units for IL-6 and explanations of all abbreviations.

All abbreviations are displayed in figure 2, are now included and corrected.

3-In Figure 2, stratifying IL-6 trends by gestational age at PPROM would add clinical context.

Thank you for your comment. Based on your recommendation we have created a new figure (Figure 2B), to demonstrate the values measured in different gestational ages.

Round 2

Reviewer 1 Report

Comments and Suggestions for Authors

-Thanks  for addressing the raised comments.

Reviewer 2 Report

Comments and Suggestions for Authors

I have no further comments for this manuscript.